# voyAGEr, a free web interface for the analysis of age-related gene expression alterations in human tissues

**Arthur L Schneider**[†‡], **Rita Martins-Silva**[†], **Alexandre Kaizeler**[†], **Nuno Saraiva-Agostinho**[§], **Nuno L Barbosa-Morais***

Instituto de Medicina Molecular João Lobo Antunes, Faculdade de Medicina, Universidade de Lisboa, Lisbon, Portugal

**\*For correspondence:**
nmorais@fm.ul.pt

[†]These authors contributed equally to this work

**Present address:** [‡]Sia Partners, 4 Rue Voltaire, 44000 Nantes, Nantes, France; [§]European Molecular Biology Laboratory, European Bioinformatics Institute, Wellcome Genome Campus, Cambridge, United Kingdom

**Competing interest:** The authors declare that no competing interests exist.

**Abstract** We herein introduce voyAGEr, an online graphical interface to explore age-related gene expression alterations in 49 human tissues. voyAGEr offers a visualisation and statistical toolkit for the finding and functional exploration of sex- and tissue-specific transcriptomic changes with age. In its conception, we developed a novel bioinformatics pipeline leveraging RNA sequencing data, from the GTEx project, encompassing more than 900 individuals. voyAGEr reveals transcriptomic signatures of the known asynchronous ageing between tissues, allowing the observation of tissue-specific age periods of major transcriptional changes, associated with alterations in different biological pathways, cellular composition, and disease conditions. Notably, voyAGEr was created to assist researchers with no expertise in bioinformatics, providing a supportive framework for elaborating, testing and refining their hypotheses on the molecular nature of human ageing and its association with pathologies, thereby also aiding in the discovery of novel therapeutic targets. voyAGEr is freely available at https://compbio.imm.medicina.ulisboa.pt/app/voyAGEr.

## eLife assessment

This work presents an **important** online platform designed to facilitate the exploration of genes and genetic pathways implicated in human aging. Leveraging a new inference methodology, the tool enables the identification and visualization of key genes and tissues impacted by aging, facilitating scientific discovery. The methods and analyses are **convincing** and will be broadly used by scientists aiming to mine human aging RNA-seq data.

## Introduction

The ageing-associated progressive loss of proper tissue homeostasis maintenance makes age a prevalent risk factor for many human pathologies, including cancer, neurodegenerative, and cardiovascular diseases (*Campisi, 2013*; *Niccoli and Partridge, 2012*; *Wyss-Coray, 2016*). A better comprehension of the molecular mechanisms of human ageing is thus required for the development and effective application of therapies targeting its associated morbidities.

Dynamic transcriptional alterations accompany most physiological processes occurring in human tissues (*López-Otín et al., 2013*). Transcriptomic analyses of tissue samples can thus provide snapshots of cellular states therein and insights into how their modifications over time impact tissue physiology. A small proportion of transcripts has indeed been shown to vary with age in tissue (*Stegeman and Weake, 2017*) and sex-specific (*Tower, 2017*; *Austad and Fischer, 2016*; *Melé et al., 2015*; *Gershoni and Pietrokovski, 2017*; *Kassam et al., 2019*; *Mayne et al., 2016*) manners. Such variations reflect dysregulations of gene expression that underlie cellular dysfunctions (*Stegeman and Weake, 2017*).

Many studies analysed the age-related changes in gene expression in rodent tissues (*Zahn et al., 2007*; *Kimmel et al., 2019*; *Benayoun et al., 2019*; *Lui et al., 2010*; *Shavlakadze et al., 2019*; *Almanzar et al., 2020*; *Schaum et al., 2020*), emphasising the role in ageing of genes related to inflammatory responses, cell cycle, or the electron transport chain. However, while it is possible to monitor the modifications in gene expression over time in those species by sequencing transcriptomes of organs of littermates at different ages, as a close surrogate of longitudinality, such studies cannot be conducted in humans for ethical reasons. Indeed, most studies aimed at profiling ageing-related gene expression changes in human tissues focused on a single tissue (e.g. muscle *Zahn et al., 2005*; *Welle et al., 2003*; *Jozsi et al., 2000*, kidney *Rodwell et al., 2004*, brain *Galatro et al., 2017*; *Olah et al., 2018*; *Berchtold et al., 2008*; *Lu et al., 2004*; *Işıldak et al., 2020*, skin *Haustead et al., 2016*; *Holzscheck et al., 2020*, blood *Nakamura et al., 2012*; *Harries et al., 2011*, liver *Thomas et al., 2002*, retina *Yoshida et al., 2002*) and/or are limited to a comparison between young and old individuals (*Welle et al., 2003*; *Jozsi et al., 2000*; *Haustead et al., 2016*; *Thomas et al., 2002*; *Yoshida et al., 2002*), failing to fully capture the changes of the tissue-specific gene expression landscape throughout ageing (*Stegeman and Weake, 2017*). A few studies were nonetheless led on more than one tissue in humans, from post-mortem samples (*Gheorghe et al., 2014*; *Yang et al., 2015*) and biopsies (*Aramillo Irizar et al., 2018*; *Glass et al., 2013*), and in mice (*Schaum et al., 2020*; *Aramillo Irizar et al., 2018*) and rats (*Yu et al., 2014*). The age-related transcriptional profiles derived therein, either from regression (*Schaum et al., 2020*; *Gheorghe et al., 2014*; *Yang et al., 2015*; *Glass et al., 2013*) or comparison between age groups (*Aramillo Irizar et al., 2018*; *Yu et al., 2014*), highlight an asynchronous ageing of tissues (discussed in *Rando and Wyss-Coray, 2021*), with some of them more affected by age-related gene expression changes associated with biological mechanisms known to be impacted by ageing such as mitochondrial activity or metabolic homeostasis. In particular, tissue-specific periods of major transcriptional changes in the fifth and eighth decades of the human lifespan have been revealed (*Gheorghe et al., 2014*), reflecting the so-called digital ageing (*Rando and Wyss-Coray, 2021*), consistent with what is observed in mice (*Almanzar et al., 2020*; *Schaum et al., 2020*). Furthermore, despite outlining the tissue specificity of the transcriptomic signatures of human ageing, some synchronisation was found between tissues like the lung, heart, and whole blood, which exhibit a co-ageing pattern (*Yang et al., 2015*). Nevertheless, as each study followed its own specific procedures, from sample collection to data processing, results from these analyses are hard to compare with one another.

Processed data from those studies have not been made easily accessible and interpretable to researchers lacking computational proficiency but aiming to use them to test their novel hypotheses. To fill this void, we have developed voyAGEr, a web application providing flexible visualisation of comprehensive functional analyses of gene expression alterations occurring in 49 human tissues with age in each biological sex. We leverage the large RNA-seq dataset from the Genotype-Tissue Expression (GTEx) project (*Lonsdale et al., 2013*), encompassing tissue samples from hundreds of donors aged from 20 to 70 years, with a pipeline for gene expression profiling with an optimised temporal resolution. voyAGEr allows us to investigate ageing from two perspectives: (i) gene-centric – how each gene's tissue-specific expression progresses with age; and (ii) tissue-centric – how tissue-specific transcriptomes change with age. Additionally, voyAGEr enables the examination of modules of co-expressed genes altered with age in four tissues (brain cortex, skeletal muscle, left ventricle of the heart, whole blood), namely their enrichment in specific cell types, biological pathways, and association with diseases. We, therefore, expect voyAGEr to become a valuable support tool for researchers aiming to uncover the molecular mechanisms underlying human ageing. Moreover, being open-source, voyAGEr can be adapted by fellow developers to be used with alternative datasets (e.g. from other species) or to incorporate other specific functionalities.

voyAGEr is freely available at https://compbio.imm.medicina.ulisboa.pt/app/voyAGEr.

## Results

voyAGEr's interactive exploration of tissue-specific gene expression landscapes in ageing is based on sequential fitting of linear models (v. Methods) to estimate, for each gene in each tissue:

i.    the *Age* effect, i.e., how the age-associated changes in gene expression evolve with age itself;
ii.   the *Sex* effect, i.e., how the differences in gene expression between sexes evolve with age;

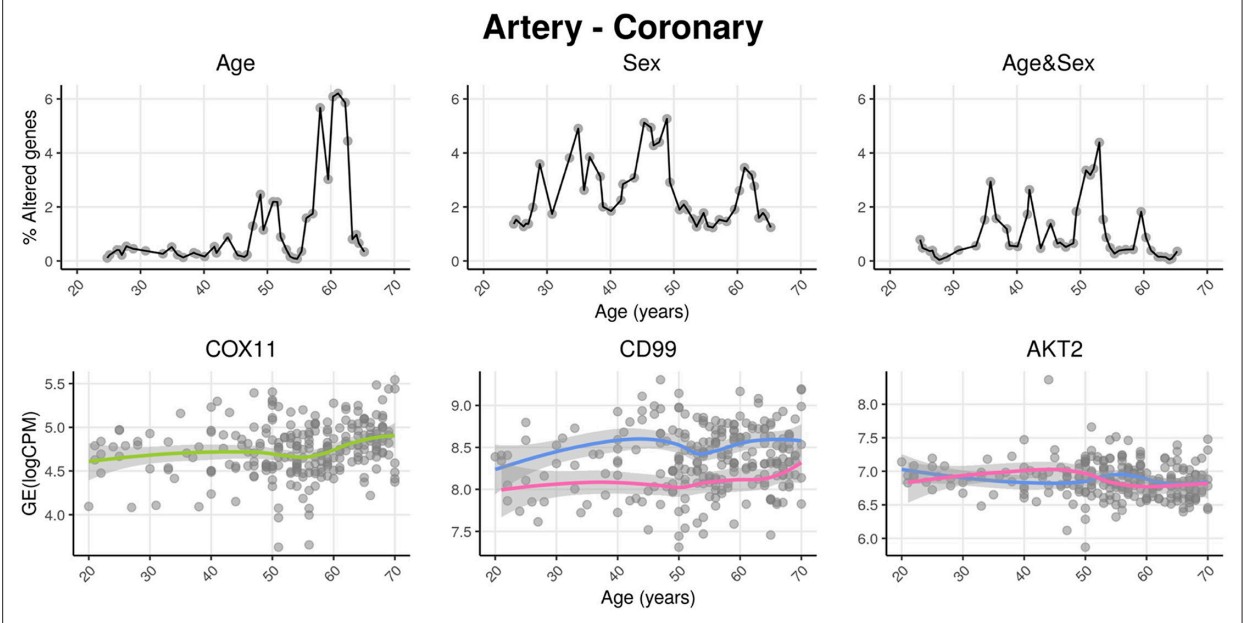

**Figure 1.** voyAGEr profiles tissue-specific age-associated changes in gene expression and their differences between sexes. For each of the 49 human tissues in genotype-tissue expression (GTEx), gene expression was linearly modelled in windows spanning 16 years centered in consecutive years of age, to estimate the effects thereon of *Age*, *Sex*, and the interaction between them, i.e., how the *Sex* effect changes with age, equivalent to how the *Age* effect differs between sexes (v. Methods). In each age window, the percentage of genes with expression significantly altered by each of those effects gives their respective transcriptomic impact (upper panels). voyAGEr thereby identifies the age periods at which major gene expression changes occur in each tissue. For example, in coronary artery: major age-related transcriptional alterations are found at around 60 years of age (upper left panel), illustrated by the behaviour of *COX11* (bottom left panel); major gene expression differences between males and females happen across the considered age range (upper centre panel), as illustrated by *CD99* (bottom centre panel); major differences between sexes in age-related gene expression alterations happen across the considered age range (upper right panel), as illustrated by *AKT2* (bottom right panel). Solid loess lines in the bottom panels (green for all donors, pink for females, blue for males). Gene expression (GE) in log$_2$ of counts per million (logCPM).

The online version of this article includes the following figure supplement(s) for figure 1:

**Figure supplement 1.** Impact of donor overlap between tissues in age-associated trends.

**Figure supplement 2.** Batch effect correction applied to lung samples.

**Figure supplement 3.** Principle of the Shifting Age Range Pipeline for Linear Modelling (ShARP-LM) method.

---

 iii. the *Age&Sex* interaction effect, i.e., how the differences between sexes of age-associated changes in gene expression evolve with age.

We named our approach Shifting Age Range Pipeline for Linear Modelling (ShARP-LM). Briefly, this method consists of performing differential gene expression (with gene expression as a function of the donors' *Age*, *Sex*, and *Age&Sex* interaction) in moving age windows spanning 16 years. By considering the percentage of genes altered in each age range, we can highlight age periods of major tissue-specific transcriptomic alterations (*Figure 1*).

### *Gene*-centric analyses of human tissue-specific expression changes across age

The progression of tissue-specific expression of a particular gene across age can be examined in voyAGEr's *Gene* tab. By entering its HGNC symbol in the Gene selector, the user has access to graphical summaries of the gene's tissue-specific expression (sub-tab *Profile*) (*Figure 2A*) and the significance of age-related changes in its expression due to the *Age*, *Sex*, and *Age&Sex* effects (sub-tab *Alteration*) (*Figure 2B*) across age. Results can be displayed in a heatmap for all tissues or in a scatter plot for a chosen individual tissue (*Figure 2C*). When the gene is studied in a single tissue, the user can graphically and statistically profile the association of the donors' sex and reported conditions (e.g. history of heart attack or pneumonia) with the gene's expression profile. A table summarizing the

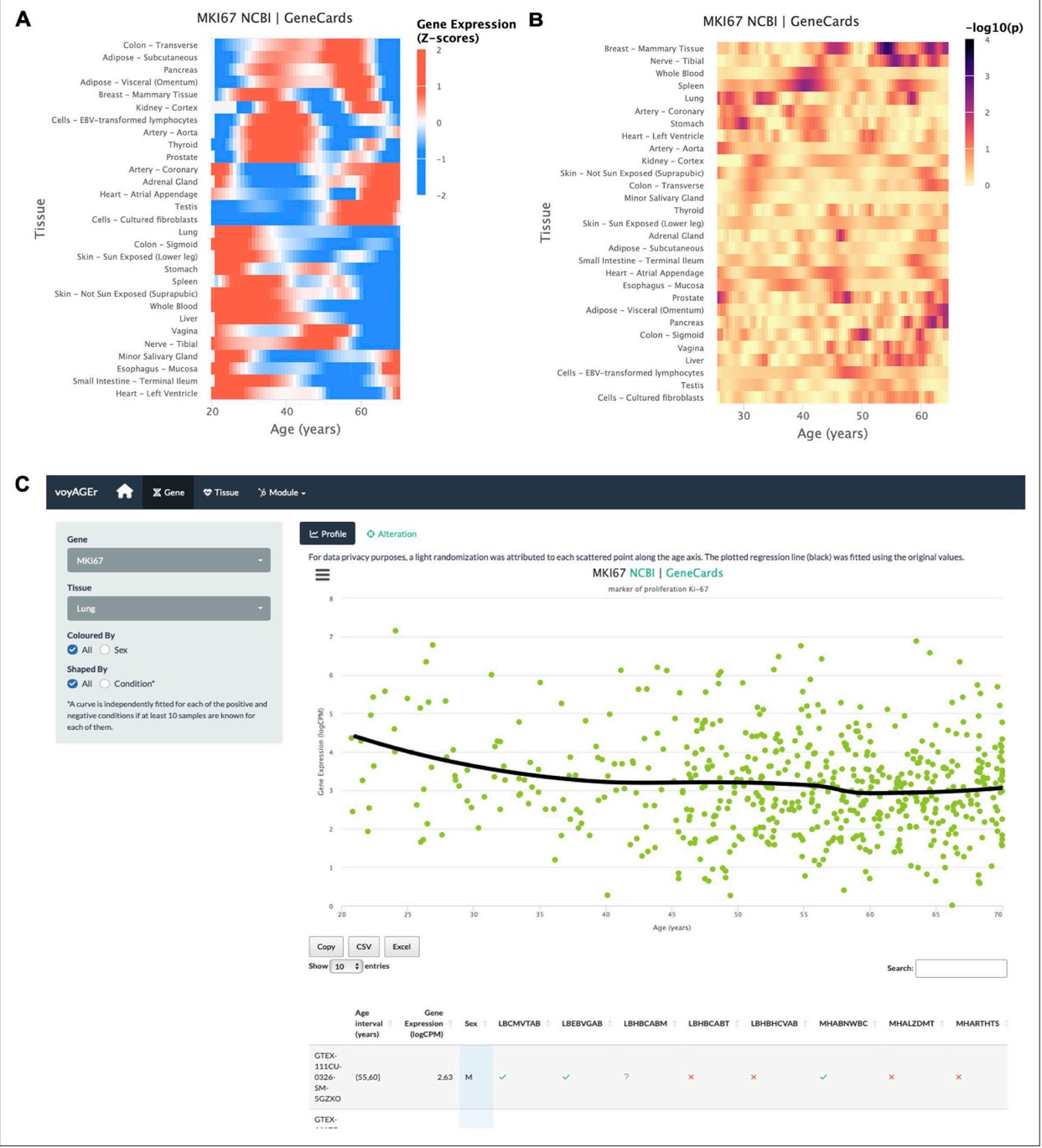

**Figure 2.** *Gene-centric* analyses of expression alterations across age. (**A**) Heatmap of *MKI67* expression across tissues over age. (**B**) Heatmap of the significance of *Age*-associated *MKI67* expression alterations over age. p-values are for the moderated t-statistics of differential gene expression associated with the Age effect (v. the ShARP-LM approach in Methods). Notably, transcriptional changes are observed in the lung (mid 20's, early 30's, and after 55). (**C**) - voyAGEr's *Gene* tab interface. *MKI67* expression in the lung is inspected. Donors' information is shown in a table and the scatter plot can be interactively adjusted according to the donors' condition of interest (*Figure 2—figure supplement 1B* ).

The online version of this article includes the following figure supplement(s) for figure 2:

**Figure supplement 1.** *voyAGEr's* Gene tab interface.

**Figure supplement 2.** Sex-specific *SALL1* and *DDX43* expression alterations over age.

**Figure supplement 3.** Impact of sex-specific genes in the interpretation of voyAGEr results.

**Figure supplement 4.** Effects of batch effect correction on gene-centric (*SFTPA2*) analysis.

donors' metadata is also shown (*Figure 2C*). The user can interactively select donors of interest on the scatter plot and further examine their information in the automatically subsetted table.

An example of a process whose molecular mechanisms are of particular interest to researchers in the ageing field is cellular senescence. Senescence is a stress-induced cell cycle arrest limiting the proliferation of potentially oncogenic cells but progressively creating an inflammatory environment in tissues as they age (*van Deursen, 2014*; *Gorgoulis et al., 2019*. *CDKN2A*, that encodes cell cycle regulatory protein p16INK4A known to accumulate in senescent cells *Gil and Peters, 2006*; *Erickson et al., 1998*), has its expression increased with age in the vast majority of tissues profiled (*Figure 2— figure supplement 1A* ). Similarly, reduced levels of proliferation markers, such as *PCNA* (*Narita et al., 2003*) and *MKI67* (*Sun and Kaufman, 2018*), can be studied as putative markers of ageing of certain tissues. These genes have their expression altered with age in the lung and display a similar expression profile (decreasing from 25 to 30 years old, constant between 35 and 50 years old and decreasing in older ages) (*Figure 2C*). However, these trends appear to vary according to the donor's history of non-metastatic cancer (*Figure 2—figure supplement 1B*), illustrating voyAGEr's use in helping to find associations between gene expression and age-related diseases.

On a different note, sex biases have been reported in the expression of *SALL1* and *DDX43* in adipose tissue and lung, respectively (*Kassam et al., 2019*). voyAGEr allows us to not only recapitulate those observations but also assess the temporal window where these changes occur (*Figure 2— figure supplement 2*).

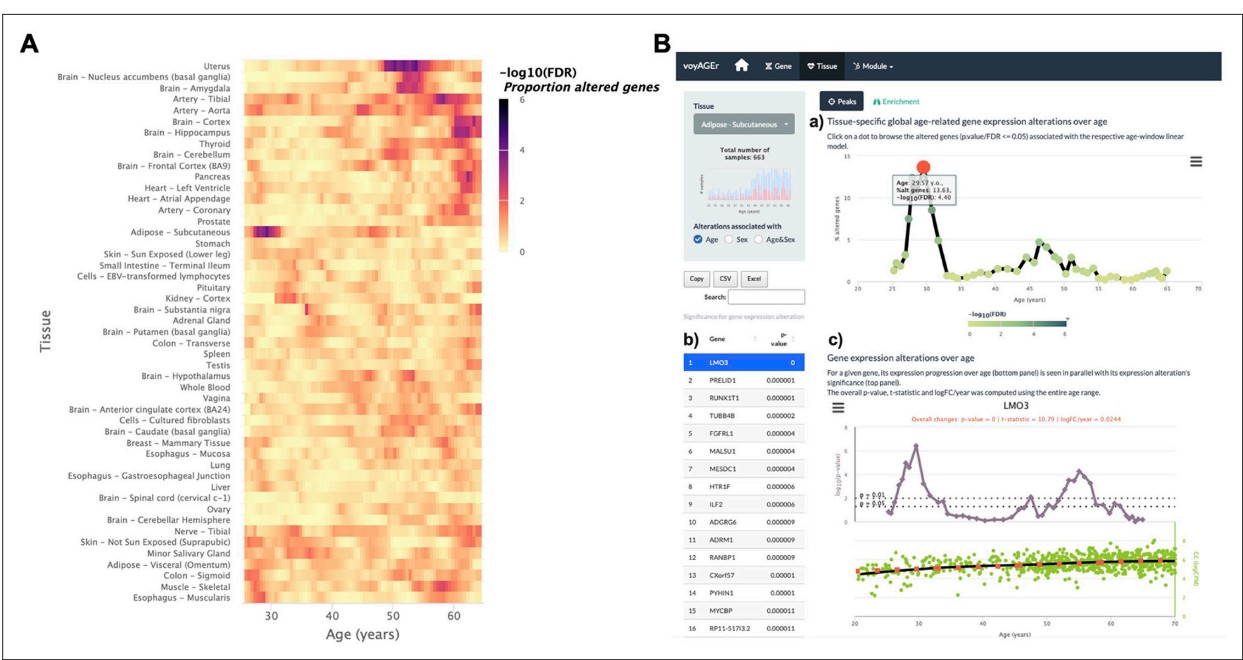

**Figure 3.** Tissue-specific assessment of gene expression changes across age. (**A**) Heatmap of significance (false discovery rate, FDR, based on random permutations of age, v. Methods) over the age of the proportion of genes with expression significantly altered with *Age* in the 49 analysed tissues. (**B**) Exploration of gene expression changes across age in Subcutaneous Adipose tissue: (**a**) Percentage of genes with significantly altered expression with *Age* over age. Two main peaks of transcriptional changes are noteworthy, a major one in the late 20s and a minor one after 45; (**b**) Clicking on the dot of a specific age (29.57 years old in plot *a*) gives access to the list of the most altered genes at that age, ordered by statistical significance of expression changes (p-value of moderated t-statistic). (**c**) Plot of expression of the chosen top gene in the table in *b* across age (bottom) in parallel with the significance of its expression alterations with *Age*. The expression of *LMO3* significantly increases at around 30 years old, concomitantly with the first peak of transcriptomic changes with *Age*.

The online version of this article includes the following figure supplement(s) for figure 3:

**Figure supplement 1.** *Transcriptomic alterations in the uterus coincident with the onset of menopause.*

**Figure supplement 2.** *Tissue-specific sex-differentiated expression.*

**Figure supplement 3.** Distributions of genotype-tissue expression (GTEx) donors' ages and its impact on the statistical power of detection of differential gene expression.

**Figure supplement 4.** Effect of downsampling in Shifting Age Range Pipeline for Linear Modelling (ShARP-LM) results.

### *Tissue*-specific assessment of gene expression changes across age

Peaks of gene expression alterations

The landscape of global tissue-specific gene expression alteration across age can be examined in voyAGEr's *Tissue* main tab. A heatmap displaying, for all tissues, the statistical significance over age (v. Methods) of the proportion of genes altered with *Age*, *Sex*, or *Age&Sex* (depending on the user's interest) is initially shown (*Figure 3A*), illustrating the aforementioned asynchronous ageing of tissues observed for humans and rodents (*Schaum et al., 2020*; *Thomas et al., 2002*; *Yoshida et al., 2002*; *Gheorghe et al., 2014*; *Yang et al., 2015*; *Aramillo Irizar et al., 2018*; *Rando and Wyss-Coray, 2021*).

The user can then spot the age periods with the most significant gene expression alterations in a selected tissue (*Figure 3B, a*), and identify the associated altered genes (*Figure 3b*). The user can also plot the expression of a given gene of interest across ages together with the significance of its expression modification with *Age*, *Sex*, or *Age&Sex* (*Figure 3B, c*).

An example of voyAGEr's capabilities is illustrated in *Figure 1—figure supplement 1*, showing substantial transcriptomic alterations in the uterus from the late forties to the early fifties, overlapping with the age distribution of menopausal onset (*Kaczmarek, 2014*), which could explain the observed molecular modifications.

It is also possible to visualise tissues with more than one period of transcriptomic changes, and to individually inspect these periods. As an example, the subcutaneous adipose tissue appears to go through two main periods of transcriptional changes with age: a major one at the late 20s (~13% of altered genes), and a minor one after 45 years (~5% of altered genes) (*Figure 3B, A*). The most altered genes in this first peak appear to have their expression modified only at this precise age period (e.g. *PRELID1*, *RUNX1T1*, *TUBB4B*, *FGFRL1,* and *MALSU1*). Similarly, mitochondrial genes (e.g. *MT-CYB*, *MT-ND4*, *MT-ATP6*, *MT-ND2*) (*Figure 3B*) appear to be the most altered genes in the second peak

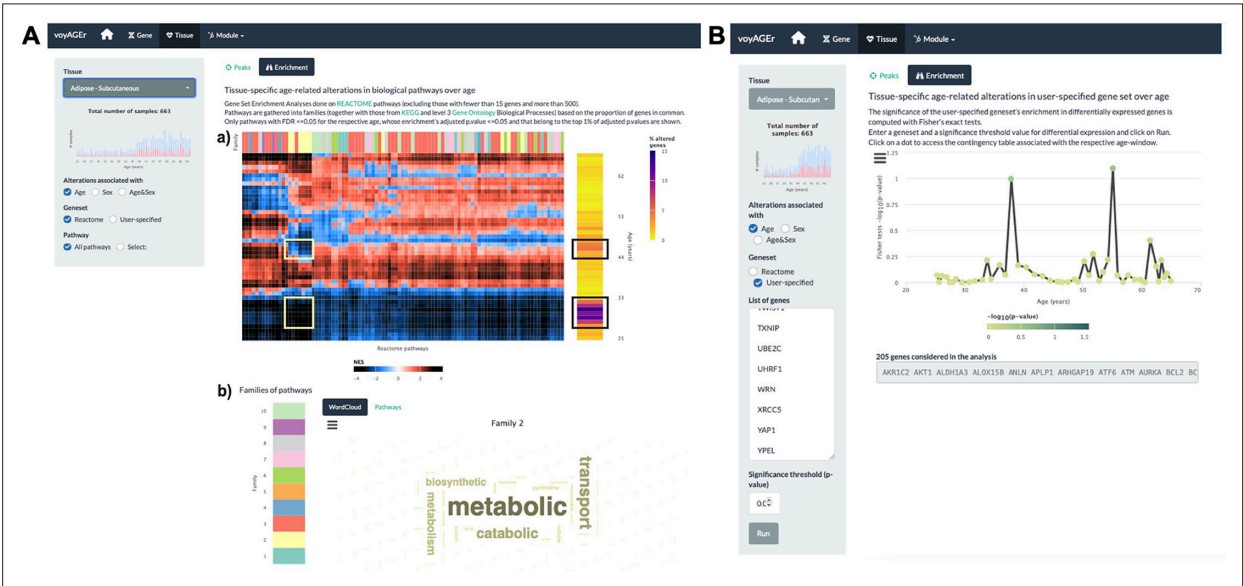

**Figure 4.** Tissue-specific assessment of pathway expression changes across age in the human Subcutaneous Adipose tissue. (**A**) Heatmap depicting the normalised enrichment scores (NES) of Reactome pathways associated with specific tissues and effects. Pathways are classified into 10 families (**a**), which can be characterised by their frequently occurring terms (**b**), providing insights into their biological functions. Only pathways significantly linked to gene expression changes in at least one age window (FDR ≤ 0.01) are displayed. Black squares indicate the two age periods with prominent transcriptional changes, while yellow squares denote pathways common to both peaks, primarily belonging to family 2. Word cloud analysis (**b**) reveals that family 2 pathways are mainly related to metabolism. (**B**) Enrichment of a user-provided gene set, given by the significance of Fisher's tests, in genes altered with *Age* throughout ageing (based on a user-defined p-value threshold). Here, the given gene set is composed of genes from Senequest *Gorgoulis et al., 2019* whose link with senescence is supported by at least four sources. In this case, there are no significant peaks.

The online version of this article includes the following figure supplement(s) for figure 4:

**Figure supplement 1.** Clustering of pathways from Reactome and Kyoto Encyclopedia of Genes and Genomes (KEGG), and level 3 Gene Ontology Biological Processes.

(*Figure 3B, C*). This particularity suggests that different sets of genes drive the periods of major transcriptional changes, which begs to assess if they reflect the activation of distinct biological processes.

## Gene set enrichment

The user can explore the biological functions of the set of genes underlying each peak of transcriptomic changes by assessing their enrichment in curated pathways from the Reactome database (*Croft et al., 2014*). voyAGEr performs Gene Set Enrichment Analysis (GSEA) (*Subramanian et al., 2005*) and the user can visualise heatmaps displaying the evolution over the age of the resulting normalised enrichment score (NES *Subramanian et al., 2005*, reflecting the degree to which a pathway is over- or under-represented in a subset of genes) for a given tissue, effect (*Age*, *Sex*, or *Age&Sex*) and Reactome pathway (all, or user-selected) (*Figure 4A*). To reduce redundancy and facilitate the understanding of their biological relevance, we clustered those pathways into families that also include Kyoto Encyclopedia of Genes and Genomes (KEGG) pathways (*Kanehisa and Goto, 2000*) and Gene Ontology (GO) Biological Processes of level 3 (*Gene Ontology Consortium, 2004*). Briefly, we clustered gene sets from the three sources based on the overlap of their genes (v. Methods), thereby creating families of highly functionally related pathways. Taking advantage of the complementary and distinct terminology in Reactome, KEGG, and GO, the user can interpret each family's broad biological function by looking at the word cloud of its most prevalent terms, and browsing the list of its associated pathways (*Figure 4A*). For example, although most pathways enriched in the two aforementioned peaks of altered genes in subcutaneous adipose tissue were different, there is an overlap of pathways related to metabolism, including various mitochondrial processes (*Figure 4A*). This highlights the importance of integrating individual gene data with pathway enrichment analysis to garner more comprehensive insights into the biological relevance of those changes.

The peaks of transcriptomic changes can also be examined for enrichment in a user-provided gene set (*Figure 4B*). As expression of senescence-related *CDKN2A* is increased in the subcutaneous adipose tissue with age (*Figure 2—figure supplement 1A*), we hypothesised that other

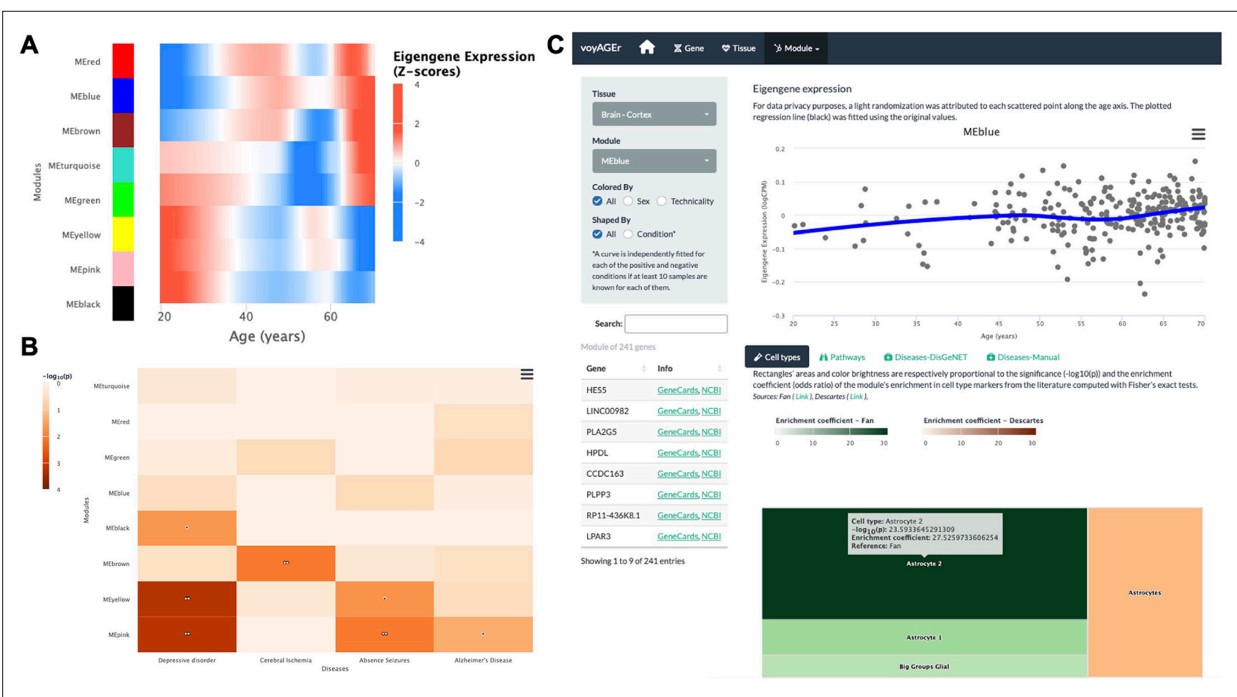

**Figure 5.** Tissue-specific assessment of age-associated progression of modules of co-expressed genes. (**A**) Heatmap of eigengene expression for all the modules of co-expressed genes in the brain cortex over age. (**B**) Heatmap of association of the modules with four selected diseases, computed with the disgenet2r package (*Piñero et al., 2019*). (**C**) Scatter plot (above) of eigengene expression over age, in all brain cortex samples, for a selected module of 241 genes (MEblue). The eigengene expression is derived from the first component of the single value decomposition of its genes' expression. This module was analysed, with Fisher's tests, for enrichment in cell types, based on markers from the literature (*Fan et al., 2018* and Descartes *Cao et al., 2020*) and found to be associated with astrocytes, as can be observed by the TreeMap below (where each rectangle's area and darkness are proportional to the significance of its association with a cell type and its colour linked to the markers' source study).

senescence-associated genes may be augmented too. Thus we used that voyAGEr functionality, using the Senequest (*Gorgoulis et al., 2019*) geneset (supported by at least four sources) to test it, observing no significant alterations (*Figure 4B*).

## Modules of co-expressed genes

voyAGEr also allows functional analyses of modules of co-expressed genes i.e. genes with highly correlated expression across samples, defined by weighted correlation network analysis (*Langfelder and Horvath, 2008*). Genes in the same module are likely to be co-regulated and share biological functions or associations with phenotypical or pathological traits (*van Dam et al., 2017*). Those modules may also act as markers of core transcriptional features of cellular activity and identity (*Kelley et al., 2018*).

Concretely, voyAGEr enables the user to visualise how the expression of modules of genes that are associated with a specific cell type, biological pathway, or disease progresses over age in a specific tissue. After selecting one tissue of interest, the user has, for each module, access to four levels of information:

1. *Expression*: its eigengene expression progression over age (*Figure 5C*);
2. *Cell types*: its enrichment in specific cell types, based on cell type signatures found in the literature (*Figure 5C*);
3. *Pathways*: its enrichment in Reactome pathways;
4. *Diseases*: its enrichment in disease markers, based on gene-disease associations from DisGeNET (*Piñero et al., 2019*; *Piñero et al., 2017*), calculated with both the disgenet2r package (*Piñero et al., 2019*) and with Fisher's tests (*Figure 5B*).

By default, for each tissue, results are displayed in the form of heatmaps of expression (centered and scaled) illustrating how all modules evolve with age (*Figure 5A*). The user also has the possibility to select a module of interest and see its eigengene progression over age in a scatter plot (*Figure 5C*), lists of its association with diseases and pathways ordered by significance, and a TreeMap for its cell type enrichments (*Figure 5C*). In the example of *Figure 5C*, the 'MEblue' module, comprising 241 genes co-expressed in the brain cortex, shows significant enrichment in astrocyte markers. The apparent increase of this module's expression with age may reflect the known age-related changes in astrocyte activation (*Palmer and Ousman, 2018*) and the relative weakening of neuronal activity ('MEyellow' and 'MEpink' modules).

As in the *Gene* tab, the user can separate donors based on their sex and associated medical conditions in the scatter plot of the eigengene expression progression. On the Pathways and Diseases-Manual tabs below the main plot, the user can also visualise the contingency table from that specific disease/pathway, on the corresponding column.

## Discussion

voyAGEr provides a framework to examine the progression of gene expression over age in several human tissues, serving as a valuable resource for the ageing research community. In particular, it helps to identify tissue-specific age periods of major transcriptomic alterations. The results of our analyses show the complexity of human biological ageing by stressing its tissue specificity (*Gheorghe et al., 2014*) and non-linear transcriptional progression throughout the lifetime, consistent with previous results from both proteomic (*Lehallier et al., 2019*) and transcriptomic (*Haustead et al., 2016*; *Gheorghe et al., 2014*) analyses. By revealing and annotating the age-specific transcriptional trends in each tissue, voyAGEr aims to assist researchers in deciphering the cellular and molecular mechanisms associated with the age-related physiological decline across the human body.

Due to the tissue-specific nature of the pre-processing steps (v. Read count data pre-processing in the Methods section), and given that most of the plotted gene expression distributions are centered and scaled by tissue, it is important to note that voyAGEr may not be always suited for direct comparisons between different tissues. For instance, it does not allow us to directly ascertain if a gene exhibits different expression levels in different tissues or if the expression of a particular gene in one tissue changes more drastically with age than in another tissue. Furthermore, we must emphasise that the majority of GTEx donors contributed samples to multiple tissues (*Figure 1—figure supplement 1* ), potentially introducing biases and confounders when comparing gene expression patterns between

tissues. Our analyses of variance (*Figure 1—figure supplement 1B* ) and downsampling to control for common donors (*Figure 1—figure supplement 1C-E*) suggest very limited global confounding between the impacts of donor and age on gene expression and that any potential cross-tissue bias not to depend much on the proportion of common donors (*Figure 1—figure supplement 1E*). However, this effect must be taken into account when comparing specific pairs of tissues (e.g. Colon – Transverse and Whole Blood, *Figure 1—figure supplement 1D*).

Additionally, voyAGEr allows us to scrutinise and visually display the tissue-specific differences in gene expression between biological sexes across ages. Biological sex is an important factor in the prevalence of ageing-associated diseases, as well as their age of onset, progression (*Tower, 2017*; *Austad and Fischer, 2016*; *Khramtsova et al., 2019*), and sex-related differences in gene expression (*Melé et al., 2015*; *Gershoni and Pietrokovski, 2017*; *Kassam et al., 2019*; *Mayne et al., 2016*). By profiling the age distribution of such differences, voyAGEr can lead to a better understanding of their influence in the aetiology of the sex specificities of human ageing. For instance, we were able to corroborate findings on the sex-differential transcriptome of adult humans by *Gershoni and Pietrokovski, 2017*, with voyAGEr emphasising its tissue-specificity and allowing to discriminate the ages at which sex-related biases appear to be more prevalent (*Figure 3—figure supplement 2*).

Nonetheless, it is essential to interpret sex chromosome-specific gene results in voyAGEr with caution. For instance, we observed elevated expression of Y-chromosome-specific *DDX3Y* in males, whilst its female expression (expected to be zero) is very low, in the range of what can be considered background noise (*Figure 2—figure supplement 3*). Its age-related alterations exhibit a distinctive peak around the age of 40 apparently driven by subtle changes in gene expression in female samples, illustrating the need for the abovementioned caution.

One of the limitations of voyAGEr is that most GTEx tissue donors had health conditions and their frequency increased with age, preventing us from defining a class of healthy individuals and identifying age-associated transcriptomic changes that could be more confidently proposed to happen independently of any disease progression. Whilst the large sample sizes and inherent biological variability among individuals, reflected in the diversity of condition combinations, are expected to mitigate significant confounding effects, voyAGEr also allows users to evaluate how tissue-specific gene expression trends vary according to the donors' diverse conditions (see *Figure 2—figure supplement 1B*).

The development of voyAGEr was accompanied by that of a pipeline, ShARP-LM, that facilitates the holistic depiction of the transcriptional landscapes of adult human tissues throughout ageing with a yearly age resolution. We take advantage of the comprehensiveness of the transcriptome collection from human tissues from the GTEx project to make our analyses a valid surrogate of a currently undoable longitudinal study. It confers our method enough statistical robustness to mitigate the inter-individual differences and deal with the non-uniform distribution of the donors' ages. Nevertheless, it is worth highlighting that the age distribution of donors does impact the statistical power for detecting transcriptional changes. Consequently, we are more likely to identify significant alterations (with p-value <0.05 in our gene-centric analyses) within age ranges that are more prevalent in our sample population, often characterised by older individuals (*Figure 3—figure supplement 3*). When downsampling to ensure a balanced age distribution, a loss of statistical power is apparent but a considerable positive correlation with the original results is maintained and a substantial number of significant alterations remain so (*Figure 3—figure supplement 4*). This limitation is likely to be overcome by the accumulation of transcriptomes of human tissues in public databases, promising a gradual increase in accuracy and age resolution with which human transcriptomic ageing can be profiled. Similarly, the expanding collection of single-cell transcriptomes in public databases is yielding improved gene markers for an increasing diversity of human cell types, enhancing the usefulness of leveraging bulk transcriptomes to study the impact of ageing on the cellular composition of human tissues, for which the co-expression module approach in voyAGEr provides a proof-of-concept.

Nonetheless, it is imperative to approach the module-based analysis with caution, as direct and literal interpretations may be misleading. For instance, it is not uncommon to observe an enrichment of 'Rheumatoid Arthritis' in modules associated with various immune cell types in anatomical locations, such as the brain cortex, where the disease does not directly manifest. If a specific module associated with a condition like 'Liver Cirrhosis' exhibits an age-related increase in the brain cortex,

of course, this does not mean such disease ever occurs within older brains. Nevertheless, we consider that the module-based approach can serve as a valuable resource for generating hypotheses.

Given its open-source nature, voyAGEr is envisaged to be a continually evolving resource, able to accommodate new data and expand its functionalities, namely by incorporating additional tissues into the modules section and integrating perturbagen data for inference of molecular causes underlying observed gene expression alterations and small molecules to target them for therapeutic purposes (*Subramanian et al., 2017*; *Saraiva-Agostinho and de Almeida, 2020*).

As an *in silico* approach with no experimental validation for its results, voyAGEr is meant to be a discovery tool, supporting biologists in the exploration of a large transcriptomic dataset, thereby generating, refining, or preliminary testing hypotheses, laying the groundwork for subsequent experimental research. It can be an entry point for projects aiming at better understanding the tissue- and sex-specific transcriptional alterations underlying human ageing, to be followed by targeted studies focusing on the functional roles of the most promising markers identified therein in the physiology of ageing. Those marker genes can contribute to the development of more robust and cross-tissue gene signatures of ageing (*de Magalhães et al., 2009*) and the expansion of age-related gene databases (*Tacutu et al., 2018*; *Craig et al., 2015*).

Moreover, the observed diverse and asynchronous changes in gene expression between tissues over the human adult life provide potentially relevant information for the design of accurate diagnostic tools and personalised therapies. On one hand, identifying those changes' association with specific disorders could have a prognostic value by enabling the identification of their onset before clinical symptoms manifest (*Aramillo Irizar et al., 2018*). On the other hand, computational screening of databases of genetic and pharmacologically induced human transcriptomic changes could help to infer their molecular causes and uncover candidate drugs to delay these effects (*Subramanian et al., 2017*; *Saraiva-Agostinho and de Almeida, 2020*; *Dönertaş et al., 2018*; *Janssens et al., 2019*).

## Methods
### Development platforms
Data analysis was performed in R (version 4.1.2) and the application developed with R package Shiny (*Chang et al., 2024*). voyAGEr's outputs are plots and tables, generated with R packages highcharter (*Kunst, 2022*) and DT (*Yihui et al., 2024*), respectively, that can easily be downloaded in standard formats (png, jpeg, and pdf for the plots; xls and csv for the tables).

voyAGEr was deployed using Docker Compose and ShinyProxy 2.6.1 in a Linux virtual machine (64 GB RAM, 16 CPU threads, and 200 GB SSD) running in our institutional computing cluster.

### Read count data pre-processing
The matrix with the RNA-seq read counts for each gene in each GTEx v8 sample was downloaded from the project's data portal (https://www.gtexportal.org/) (*Lonsdale et al., 2013*). From the 54 tissues available from GTEx v8, five were discarded (kidney medulla, fallopian tube, bladder, ectocervix, endocervix) due to low (<50) numbers of samples.

Read count data for each tissue were then pre-processed separately. We started by filtering out genes deemed uninformative due to their very low expression across samples: only genes with at least one CPM in at least 40% of the samples were kept for analysis (the number of genes analysed for each tissue can be found in *Supplementary file 1*; *Robinson et al., 2010*). Read counts for those kept genes were used to calculate normalisation factors to scale the raw library sizes, using function calcNormFactors from edgeR (*Robinson et al., 2010*) that implements the trimmed mean of M-values (*Robinson and Oshlack, 2010*). Read counts were subsequently normalised and log-transformed with the voom function (*Law et al., 2014*) from package limma (*Ritchie et al., 2015*).

However, it is well-established that batch effects, which may stem from variations in sample treatment prior to RNA-seq library preparation, can introduce spurious gene expression differences between samples and result in confounding factors (*García-Pérez et al., 2023*). We, therefore, conducted an impartial and systematic search for potential batch effects. Firstly, we performed a principal component analysis of gene expression for each tissue, using the prcomp package. We quantified the relation between each condition associated with every sample [According to the annotated variables for the dbGaP Study Accession phs000424.v8.p2] and the first two principal components,

by computing Spearman correlations (for numerical conditions), t-tests (for binary categorical conditions), or analysis of variance (ANOVA) tests (for variables with more than two possible values, and, in the case of numerical variables, fewer than 15 unique values). Conditions that surpassed defined empirical thresholds (p-value <0.05, Spearman correlation >0.3, t-test >10, and ANOVA >20) were flagged as potential batch effects. Except in brain tissues, the COHORT variable (i.e. whether the samples were collected from organ donors or *post-mortem*) appeared to be the main batch effect, with ripple effects on numerous other related conditions (*Figure 1—figure supplement 2*). Moreover, SMRIN (sample's RNA integrity number), DTHHRDY (death classification based on the 4-point Hardy Scale), and MHSMKYRS (smoke years) consistently emerged as conditions associated with the primary axes of variance. The number of genes detected in each sample, determined by the filtration step described above, was also identified as a significant contributor to the primary data variance. We, therefore, corrected for these five conditions, on a tissue-by-tissue basis, by adapting the remove-BatchEffect function from the limma package (*Ritchie et al., 2015*). Specifically, we employed linear models to estimate the contributions to gene expression of each of those factors and subtracted such contributions from the original logCPM matrix. To ensure the biological interpretability of the results, we offset the resulting values to the minimum value in the non-corrected matrix. To prevent sample loss due to missing values for the aforementioned five conditions and since the number of missing values was relatively low compared to the total number of samples, imputation was carried out using the mice package (*Buuren and Groothuis-Oudshoorn, 2011*).

The resulting matrix of logCPM-corrected values was used for all downstream analyses. As an illustrative example of the importance of batch removal, the expression of surfactant factor *SFTPA2* was found to be associated with donors on a ventilator (*McCall et al., 2016*). Without batch correction, *SFTPA2* expression would have been associated with age due to the higher prevalence of such cases among older individuals (*Figure 2—figure supplement 4*).

## ShARP-LM

To model the changes in gene expression with age, we developed the Shifting Age Range Pipeline for Linear Modelling (ShARP-LM). For each tissue, we fitted linear models to the gene expression of samples from donors with ages within windows with a range of 16 years shifted through consecutive years of age (i.e. in a sliding window with window size = 16 and step size = 1 years of age). This was the minimum age span needed to guarantee the presence of more than one sample per window, across all considered tissues. As samples at the ends of the dataset's age range would be thereby involved in fewer linear models, we made the window size gradually increase from 11 to 16 years when starting from the 'youngest' samples and decrease from 16 to 11 years when reaching the 'oldest' (*Figure 1—figure supplement 3*).

Function lm from limma was used to fit the following linear model for gene expression (GE):

$$\text{GE} \sim \text{E}_0 + \alpha.\text{Age} + \beta.\text{Sex} + \chi.\text{Age} + \text{Sex} + \varepsilon$$

For each gene, α, β, and γ are the coefficients to be estimated for their respective hypothesised effects. For each sample, *Age* in years and *Sex* in binary were centered and *Age&Sex* interaction was given by their product. The coefficients $\text{E}_0$ and ε are thus the expression of the average sample (i.e. with average sex and average age) and the error term.

For each gene in each model (i.e. each age window in each tissue), we retrieved the t-statistics of differential expression associated with the three relevant variables and their respective p-values. We considered the average age of the samples' donors within the age window as the representative age of the observed expression changes.

In summary, for a given tissue and variable (*Age*, *Sex*, and *Age&Sex*), ShARP-LM yields t-statistics and p-values over age for all genes, reflecting the magnitude and significance of the changes in their expression throughout adult life.

## *Gene*-centric visualisation of tissue-specific expression changes across age

For visualisation purposes, the trend of each gene's expression progression over age in each tissue was derived through Local Polynomial Regression Fitting, using R function loess on logCPM values (*Figures 1–3B, C*). For summarizing in a heatmap a given gene's expression across age in multiple

tissues (*Figure 2A*) or the expression of several genes across age in a specific tissue, each gene's regression values are centered and scaled, using R function scale.

For summarizing in a heatmap the significance of a given gene's expression changes over age in multiple tissues (*Figure 2B*), cubic smoothing splines are fitted to $-\log_{10}(p)$, with p being the t-statistic's p-value, with R function smooth.spline.

## Tissue-specific quantification of global transcriptomic alterations across age

To assess the global transcriptomic impact of each of the three modelled effects in each tissue across age, we analysed the progression over age (i.e. over consecutive linear models) of the percentage of genes whose expression is significantly altered (t-statistic's p-value ≤0.01) by each effect (*Figure 3B*). To evaluate the significance of each percentage and assess if high percentages can be confidently associated with major transcriptomic alterations, we controlled for their false discovery rate (FDR) by randomly permuting the samples' ages and sexes within each age window fifty thousand times and performing ShARP-LM on each randomised dataset. We were then able to associate an FDR to each percentage of altered genes by comparing it with the distribution of those randomly generated (*Figure 3B, A*).

We also applied a linear model across the entire age range, thereby providing users with more insight and supporting evidence into how a specific gene changes with age. For visualisation purposes, we incorporated a dashed orange line, with the logFC per year for the Age effect as slope, in the respective scatter plots (*Figure 3B, C*). We depict the Sex effect therein by prominent dots on the average samples, with pink and blue denoting females and males, respectively.

## GSEA

For each *Peak* of significant gene expression modifications, we performed GSEA (*Subramanian et al., 2005*) on the ordered (from the most positive to the most negative) t-statistics of differential expression for the respective tissue and age, using R package fgsea (*Korotkevich et al., 2021*) and the Reactome database (*Croft et al., 2014*). We extracted the GSEA normalised enrichment score (NES), which represents the degree to which a certain gene set is overrepresented at the extreme ends of the ranked list of genes. A positive NES corresponds to the gene set's overrepresentation amongst up-regulated genes within the age window, whereas a negative NES signifies its overrepresentation amongst down-regulated genes. The NES for each pathway was used in subsequent analyses as a metric of its up- or down-regulation in the *Peak*. The resulting NES for each pathway was used in subsequent analyses as a metric of its over- or down-representation in the *Peak*.

To optimise computational efficiency and minimise redundancy in the analysed pathways, we only considered pathways containing a minimum of 15 genes and up to 500 genes, as suggested in the GSEA User Guide (*GSEA-MSIGDB, 2023*). For the sake of clarity in voyAGEr's visual representations, we only included pathways with a p-adjusted value less than or equal to 0.05, further narrowing it down to pathways within the top 1% of p-adjusted values. Additionally, we exclusively featured pathways with at least one significant age Peak (FDR ≤ 0.05), as illustrated in *Figure 4A*.

## Families of pathways

To reduce pathway redundancy and facilitate the assessment of their biological relevance in the results' interpretation, we created an unifying representation of pathways from Reactome (*Croft et al., 2014*), KEGG (*Kanehisa and Goto, 2000*), and level 3 Biological Processes from GO (*Gene Ontology Consortium, 2004*), by adapting a published pathway clustering approach (*Chen et al., 2014*) to integrate them into families.

The approach relies on the definition of a hierarchy of pathways based on the number of genes they have in common. For each two pathways $P_i$ and $P_j$, respectively containing sets of genes $G_i$ and $G_j$, we computed their overlap index (OI) (*Chen et al., 2014*), defined as follows:

$$OI_{i,j} = |G_i \cap G_j| / \min(|G_i|, |G_j|)$$

Where $|G_i|$ is the number of genes in set $G_i$ and $|G_i \cap G_j|$ is the number of genes in common between $G_i$ and $G_j$. $OI_{ij} = 1$ would, therefore, indicate that $P_i$ and $P_j$ are identical in gene composition or that one is a subset of the other. On the contrary, $OI_{ij} = 0$ would mean that $P_i$ and $P_j$ have no genes in common.

To ease the computational work, we removed from the analysis pathways that are subsets of larger pathways (i.e. each pathway whose genes are all present in another pathway).

From the $OI_{ij}$ matrix, from which each row is a vector with the gene overlaps of pathway $i$ with each of the pathways, we computed the matrix of Pearson's correlation between all pathways' overlap indexes with R function cor. That matrix was finally transformed into Euclidean distances with R function dist, allowing for pathways to be subsequently clustered with the complete linkage method with R function hclust. The final dendrogram was empirically cut into 10 clusters (*Figure 4—figure supplement 1*). Pathways that were initially excluded from the computation for being subsets of others were added to the clusters of their respective *parent* pathways. Each *daughter* pathway with more than one parent was assigned to the cluster of the parent with the smallest number of genes, thereby maximizing the daughter-parent similarity. The data.table R package, for fast handling of large matrices, was used in this analysis (*Robertson et al., 1992*).

## Gene co-expression modules

Gene co-expression modules were defined with R package WGCNA (*Langfelder and Horvath, 2008*). For each considered tissue, the process began with the identification of a set of informative genes that exhibit high variability across samples (referred to as variable A), thus improving module definition. Next, we calculated the bicorrelation matrix for the expression of all selected genes with the bicor function. We then applied soft thresholding by raising all correlation values to the power of β, accentuating stronger correlations. The value of $\beta=12$ was chosen in accordance with the WGCNA FAQs (*Langfelder and Horvath, 2024*), and after confirmation of a free-scale topology using the pickSoftThreshold function. We generated the dissimilarity matrix by subtracting the output of the TOMsimilarity function from 1. Gene co-expression modules were then defined using a static tree-cutting algorithm, implemented via the cutreeStaticColor function, requiring as parameters a minimum number of genes per module (referred to as variable B) and the tree-cutting height (variable C).

The empirical determination of parameters A to C was guided by the following principles: (i) maximizing cell type signature enrichment, (ii) minimizing the number of genes per module, and (iii) ensuring that the modules' eigengenes exhibited age-related variability. Different combinations of these variables were exhaustively tested. To maintain biological relevance, modules consisting of non-assigned genes or genes lacking substantial supporting evidence, such as pseudogenes, were excluded.

Maximizing cell type enrichment in the modules, with a focus on known markers for specific cell types, has previously been proven successful in unveiling core transcriptional features of cell types in the human central nervous system (*van Dam et al., 2017*). Cell-type enrichment analysis relied on Fisher tests, providing odds enrichment scores and significance values (p-values). This involved comparing module genes with the signature, considering as background of genes those included in the module definition for each tissue. We prioritised modules of genes for which we obtained at least one significant result for each cell type (odds ratio (OR) >1 and p-value <0.05). The cell type signatures employed in this analysis were sourced from MSigDB's C8 collection (*Subramanian et al., 2005*), and in the case of Whole Blood, additionally from LM22 (*Chen et al., 2018*).

Specific variance thresholds (variable A) were employed: 0.5, 0.4, 0.35, and 0.9 for Brain – Cortex, Muscle – Skeletal, Heart – Left Ventricle, and Whole Blood, respectively. The minimum number of genes per module (variable B) was set at 15, 20, 20, and 15, respectively. Tree-cutting heights (variable C) of 0.95, 0.98, 0.99, and 0.97 were respectively used.

Each module is characterised by a set of genes and an eigengene, represented by the first principal component obtained through singular-value decomposition of the module's gene expressions. Subsequently, voyAGEr facilitates an evaluation of cell type enrichment, as described earlier, and enrichment in biological pathways and diseases. The enrichment of modules in cell types, Reactome pathways, and diseases (extracted from DisGeNET database version 7.0; *Piñero et al., 2019*; *Piñero et al., 2017*) was quantified using Fisher's tests. For disease enrichment, the function disease_enrichment from the disgenet2r package (*Piñero et al., 2019*) was employed, utilising their curated set of diseases. The significance of these enrichments was determined through p-value/FDR adjustment using Benjamini-Hochberg correction. For visual clarity, only pathways and diseases displaying significant enrichment (p≤0.05) in at least one module were considered.

The four tissues (Brain - Cortex, Muscle - Skeletal, Heart - Left Ventricle, and Whole Blood) covered by the Module section of voyAGEr were selected due to their relatively high sample sizes and availability of comprehensive cell type signatures. The increasing availability of human tissue scRNA-seq datasets (e.g., through the Human Cell Atlas; *Regev et al., 2017*) will allow future updates of voyAGEr to encompass a wider range of tissues.

## Data and code availability

Processed GTEx v8 RNA-seq data (read count tables) were downloaded from the project's data portal (https://www.gtexportal.org/). Donor metadata were obtained from dbGaP - database of Genotypes and Phenotypes (Accession phs000424.v9.p2 project ID 13661). voyAGEr's output tables can be directly downloaded in standard xls and csv formats. The complete source code for voyAGEr (v2.0.0 for the analyses reported herein), including pre-processing and Shiny app, can be accessed on GitHub at the following repository: https://github.com/DiseaseTranscriptomicsLab/voyAGEr.

## Acknowledgements

We thank iMM colleagues Joana Neves and Luísa Lopes, as well as all members from the Disease Transcriptomics Lab, for providing valuable feedback on the manuscript. We also thank the very knowledgeable anonymous reviewers designated by *eLife* for their constructive and insightful suggestions and criticisms to the first version of this manuscript; their feedback greatly contributed to much improved versions of both the article and voyAGEr itself. This work was supported by European Molecular Biology Organization (EMBO Installation Grant 3057), FCT - Fundação para a Ciência e a Tecnologia IP. (FCT Investigator Starting Grant IF/00595/2014, CEEC Individual Assistant Researcher contract CEECIND/00436/2018, PhD Studentships SFRH/BD/131312/2017, UI/BD/153368/2022 and UI/BD/153363/2022, projects LA/P/0082/2020 and UIDP/50005/2020), European Union (EU) Horizon 2020 Research and Innovation Programme (RiboMed 857119), and GenomePT project (POCI-01–0145-FEDER-022184), supported by COMPETE 2020 – Operational Programme for Competitiveness and Internationalization (POCI), Lisboa Portugal Regional Operational Programme (Lisboa2020), Algarve Portugal Regional Operational Programme (CRESC Algarve2020), under the PORTUGAL 2020 Partnership Agreement, through the European Regional Development Fund (ERDF), and by FCT - Fundação para a Ciência e a Tecnologia.

## Additional information

### Funding

| Funder | Grant reference number | Author |
|---|---|---|
| European Molecular Biology Organization | Installation Grant 3057 | Nuno L Barbosa-Morais |
| Fundação para a Ciência e a Tecnologia | FCT Investigator Starting Grant IF/00595/2014 | Nuno L Barbosa-Morais |
| Fundação para a Ciência e a Tecnologia | CEEC Individual Assistant Researcher contract CEECIND/00436/2018 | Nuno L Barbosa-Morais |
| Fundação para a Ciência e a Tecnologia | PhD Studentship SFRH/BD/131312/2017 | Nuno Saraiva-Agostinho |
| Fundação para a Ciência e a Tecnologia | PhD Studentship UI/BD/153368/2022 | Rita Martins-Silva |
| Fundação para a Ciência e a Tecnologia | PhD Studentship UI/BD/153363/2022 | Alexandre Kaizeler |
| Fundação para a Ciência e a Tecnologia | Project LA/P/0082/2020 | Arthur L Schneider Rita Martins-Silva Alexandre Kaizeler Nuno Saraiva-Agostinho Nuno L Barbosa-Morais |

| Funder | Grant reference number | Author |
|---|---|---|
| Fundação para a Ciência e a Tecnologia | Project UIDP/50005/2020 | Arthur L Schneider<br>Rita Martins-Silva<br>Alexandre Kaizeler<br>Nuno Saraiva-Agostinho<br>Nuno L Barbosa-Morais |
| Horizon 2020 Framework Programme | Twinning grant RiboMed 857119 | Nuno L Barbosa-Morais |
| Fundação para a Ciência e a Tecnologia | GenomePT project POCI-01-0145-FEDER-022184 | Arthur L Schneider<br>Nuno L Barbosa-Morais |
| European Regional Development Fund | GenomePT project POCI-01-0145-FEDER-022184 | Arthur L Schneider<br>Nuno L Barbosa-Morais |

The funders had no role in study design, data collection and interpretation, or the decision to submit the work for publication.

## Author contributions

Arthur L Schneider, Conceptualization, Resources, Data curation, Software, Formal analysis, Investigation, Visualization, Methodology, Writing – original draft; Rita Martins-Silva, Alexandre Kaizeler, Resources, Data curation, Software, Formal analysis, Validation, Investigation, Visualization, Methodology, Writing - review and editing; Nuno Saraiva-Agostinho, Resources, Software; Nuno L Barbosa-Morais, Conceptualization, Supervision, Funding acquisition, Investigation, Methodology, Writing – original draft, Project administration, Writing - review and editing

## Author ORCIDs

Rita Martins-Silva ⬚ http://orcid.org/0000-0002-1067-7993
Alexandre Kaizeler ⬚ http://orcid.org/0000-0002-9117-6073
Nuno Saraiva-Agostinho ⬚ https://orcid.org/0000-0002-5549-105X
Nuno L Barbosa-Morais ⬚ http://orcid.org/0000-0002-1215-0538

Reviewer #1 (Public Review): https://doi.org/10.7554/eLife.88623.3.sa1
Reviewer #2 (Public Review): https://doi.org/10.7554/eLife.88623.3.sa2
Reviewer #3 (Public Review): https://doi.org/10.7554/eLife.88623.3.sa3
Author Response https://doi.org/10.7554/eLife.88623.3.sa4

# Additional files

## Supplementary files

• Supplementary file 1. Number of studied genes per tissue. For each tissue, we kept the genes with at least one CPM of expression in at least 40% of the samples.

• MDAR checklist

## Data availability

No data has been generated for this manuscript. The data used for the analyses described in this manuscript were obtained from the GTEx Portal (https://www.gtexportal.org/) and dbGaP accession number phs000424.v9.p2 on 06/12/2023. The complete source code for data processing and Shiny interface is available on GitHub: https://github.com/DiseaseTranscriptomicsLab/voyAGEr.

The following previously published dataset was used:

| Author(s) | Year | Dataset title | Dataset URL | Database and Identifier |
|---|---|---|---|---|
| The GTEx Consortium | 2022 | Common Fund (CF) Genotype-Tissue Expression Project (GTEx) | https://www.ncbi.nlm.nih.gov/projects/gap/cgi-bin/study.cgi?study_id=phs000424.v9.p2 | NCBI Gene Expression Omnibus, phs000424.v9.p2 |

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
