## [Editor Report · eLife assessment]

This work presents an **important** online platform designed to facilitate the exploration of genes and genetic pathways implicated in human aging. Leveraging a new inference methodology, the tool enables the identification and visualization of key genes and tissues impacted by aging, facilitating scientific discovery. The methods and analyses are **convincing** and will be broadly used by scientists aiming to mine human aging RNA-seq data.

---

## [Referee Report · Reviewer #1 (Public Review)]

This fascinating paper by A.L. Schneider et al. describes voyAGEr, a shiny-based interface for easy exploration of the GTEx dataset by non- or novice programmers. Importantly, voyAGEr is open source and available from github, which could greatly accelerate additional development and further uses of this interesting tool.

The authors developed a pipeline for modeling age-related changes in gene expression in the GTEx data called ShARP-LM, fitting a linear model for age, sex and age&sex interaction terms. This pipeline underlies the later analyses that can be applied within voyAGEr. These analyses are labeled by tissue so that users can easily begin a query based on a tissue or a gene of possible interest.

voyAGEr implements many kinds of interesting R-based tools such as pathway overrepresentation analysis and gene co-expression module analysis, in a way that akes these approaches accessible to non-bioinformaticist aging researchers.

As the tidal wave of publicly available large, high-dimensional datasets such as transcriptomes continues to grow exponentially, the usefulness of tools such as voyAGEr will only increase. While test users may be able to imagine features or refinements they wish were already present, due to the open source approach they or anyone else including but not limited to the present authors can implement additional features in the future. I look forward to using this tool and to staying abreast of its future development.

Overall, this study describes a new tool of interest to the field. The manuscript is clearly written overall, with a few minor suggested corrections, as noted below. The figures and supplementary information are all clear and all add to the manuscript.

---

## [Referee Report · Reviewer #2 (Public Review)]

The purpose of this study is to develop a tool that serves as a starting point for investigating and uncovering genes and pathways associated with aging. The tool utilizes information from the GTEx public database, which contains post-mortem human data. It focuses on identifying age-related gene expression changes across different age range, biological sexes, and medical histories, with a focus on specific tissues.

Additionally, the authors envision the platform as continuously evolving, with ongoing development and expansion to include new data and features, ensuring it remains a cutting-edge resource for researchers studying aging.

voyAGEr presents a tool for exploring gene expression changes across multiple tissues in the context of aging. One of the main strengths of the tool is its intuitive and user-friendly interface, which allows for easy navigation and exploration of gene expression patterns for biologists. Users can explore changes in gene expression of single genes across multiple tissues, enabling them to identify genes of interest that can be further investigated.

A particularly noteworthy strength of the tool is its ability to show tissue-specific gene expression patterns. This feature is essential for elucidating the paradigm of tissue-specific asynchronous aging and provides a unique and valuable resource for the aging community.

However, the choice of the R shiny platform for visualization may not be the most conducive to extensibility and open-source collaboration, owing to its lack of modularity. Alternatives like Flask or FastAPI, which are more production-oriented, could be more appropriate. Additionally, despite using preprocessed data and functioning primarily as a visualization platform, the tool occasionally experiences lag, indicating room for performance improvement. These aspects are worth considering for future versions of the tool.

Overall, voyAGEr offers an entry point for further investigation of genes involved in aging, and its ability to show tissue-specific gene expression patterns provides a unique and valuable resource for the scientific community.

Finally, the tool is complemented by a comprehensive tutorial that elucidates each functionality and includes examples. The authors have shared the code for preprocessing and the tool itself. They also acknowledge the limitations of the statistical inference tests and their interpretation in the manuscript, contributing to its transparency.

---

## [Referee Report · Reviewer #3 (Public Review)]

In their manuscript, Schneider et al. aim to develop voyAGEr, a web-based tool that enables the exploration of gene expression changes over age in a tissue- and sex-specific manner. The authors achieved this goal by calculating the significance of gene expression alterations within a sliding window, using their unique algorithm, Shifting Age Range Pipeline for Linear Modelling (ShARP-LM), as well as tissue-level summaries that calculated the significance of the proportion of differentially expressed genes by the windows and calculated enrichments of pathways for showing biological relevance. Furthermore, the authors examined the enrichment of cell types, pathways, and diseases by defining the co-expressed gene modules in four selected tissues. Although their algorithm ShARP-LM has limited statistical power due to its calculation within a 16-year window, the voyAGEr was developed as a discovery tool, giving researchers easy access to the vast amount of transcriptome data from the GTEx project. Overall, the research design is unique and well-performed in simulating age-dependent changes in gene expression. The interesting results provide useful resources for the field of human genetics of aging.

---

## [Author Response]

The following is the authors’ response to the original reviews.

We are pleased to send you a revised version of our manuscript entitled “voyAGEr: free web interface for the analysis of age-related gene expression alterations in human tissues” and the associated shiny web app, in which we incorporate the referees’ feedback. We would like to express our gratitude for their time and valuable insights, which have contributed to the improvement of our work. We appreciate the rigorous evaluation process that eLife maintains.

In this letter, we address each of the reviewers' comments and concerns, point-by-point, offering detailed responses and clarifications. We have made several revisions to our manuscript following their recommendations.

We must note that the revised version of the manuscript has two novel joint first authors, Rita Martins-Silva and Alexandre Kaizeler, who performed all the requested reanalyses, given that the initial first author, Arthur Schneider, already left our lab. We must also point to the following minor unsolicited improvements we took the opportunity to make:

Added a comprehensive tutorial to the GitHub repository on how to navigate through voyAGEr’s features.Implemented sample randomisation in the scatter plots depicting gene expression across the age axis to ensure data privacy.Implemented minor adjustments within the web app to enhance user comprehension and clarity when visualizing the data.Improved clarity of the methodological sections.

**Reviewer 1**
(1.1) While this may be obvious to others for some reason that escaped me, I was unsure what was the basis for the authors' choice of 16 years as the very specific sliding window size. If I'm not alone in this, it might add clarity for other readers and users if this parameter choice were explained and justified more explicitly.

We apologise for our omission in providing the rationale behind our choice in the previous version. We chose 16 years as our sliding window size because this was the minimum needed to guarantee the presence of more than one sample per window, across all the tissues considered in the study (Figure R1 below).

We added the following sentence to the manuscript (v. Methods, ShARP-LM):

“This was the minimum age span needed to guarantee the presence of more than one sample per window, across all considered tissues.”

(1.2) "In particular, tissue-specific periods of major transcriptional changes in the fifth and eighth decades of human lifespan have been revealed, reflecting the so-called digital aging and consistently with what is observed in mice" here I think that "consistently" should be "consistent".

We thank the reviewer for the comment and following the suggestion, we have revised 'Consistently' to 'consistent' as it is the correct usage in our sentence.

(1.3) "On a different note, sex biases have been reported in for the expression of SALL1 and KAL1 in adipose tissue and lung, respectively." Here I think that "in for" should be "in".

As recommended by the reviewer, we have replaced ‘in for’ for ‘in’. As we substituted KAL1, the current sentence now stands as “On a different note, sex biases have been reported in the expression of SALL1 and DDX43 in adipose tissue and lung, respectively”.

(1.4) "We downloaded the matrix with the RNA-seq read counts for each gene in each GTEx v7 sample from the project's data portal (https://www.gtexportal.org/)." In my pdf manuscript this hyperlink appears to be broken.

We appreciate the reviewer's attention to the broken link, and we have rectified the issue. The link should now be fully operational, effectively directing users to the GTEx Portal.

(1.5) Under methods, I might suggest "Development platform" or "Development platforms" over "Development's platform" as a heading.

We have modified the heading of this section in the methods to 'Development Platforms', as we believe it better reflects the information conveyed.

**Reviewer 2**
(2.1) In this tool/resource paper, it is crucial that the data used is up-to-date to provide the most comprehensive and relevant information to users. However, the authors utilized GTEx v7, which is an outdated (2016) version of the dataset. It is worth noting that GTEx v8 includes over 940 individuals, representing a 35% increase in individuals, and a 50% increase in the total number of samples. The authors should check the newer versions of GTEx and update the data.

When the development of the voyAGEr web application began, GTEx version 7 was the most up to date. Nevertheless, we agree that the version 8 offers a notably more extensive dataset, encompassing a larger number of individuals, samples, and introducing new tissues. Consequently, we have updated our application to incorporate the data from GTEx version 8.

(2.2) The authors did not address any correction for batch effects or RNA integrity numbers, which are known to affect transcriptome profiles. For instance, our analysis of GTEx v8 Cortex tissue revealed that after filtering out lowly expressed genes, in the same way authors did, PC1 (which accounts for 24% of the variation) had a Spearman's correlation value of 0.48 (p<6.1e-16) with RNA integrity number.

We acknowledge the validity of the reviewer’s comment and appreciate the importance of such corrections to enhancing data interpretation. In response, we conducted a thorough unbiased investigation into potential batch effects, with the COHORT variable emerging as the primary driver of those observed across most tissues. Furthermore, SMRIN (as the reviewer pointed), DTHHRDY, MHSMKYRS and the number of detected genes in each sample were consistently associated with the primary sources of variation. As a result, we implemented batch effect correction for those five conditions, in a tissue-specific manner.

We provide a detailed explanation of the batch effect correction methodology and its importance in the biological interpretation of results in the Methods section, specifically under "Read count data pre-processing". Additionally, we have included two new supplementary figures, Sup. Figures 7 and 8, to illustrate a batch effect example in lung tissue and emphasise the critical role of this correction in data interpretation.

(2.3) The data analyzed in the GTEx dataset is not filtered or corrected for the cause of death, which can range from violent and sudden deaths to slow deaths or cases requiring a ventilator. As a result, the data may not accurately represent healthy aging profiles but rather reflect changes in the transcriptome specific to certain diseases due to the age-related increase in disease risk. While the authors do acknowledge this limitation in the discussion, stating that it is not a healthy cohort and disease-specific analysis is not feasible due to the limited number of samples, it would be useful for users to have the option to analyze only cases of fast death, excluding ventilator cases and deaths due to disease. This is typically how GTEx data is utilized in aging studies. Alternatively, the authors should consider including the "cause of death" variable in the model.

This comment is closely related to the prior discussion (point 2.2). Notably, two of the covariates selected for batch effect correction, namely, DTHHRDY (Death classification based on the 4-point Hardy Scale1) and COHORT (indicating whether the participant was a postmortem, organ, or surgical donor1), have a direct relevance to this issue, i.e., both relate to the cause of death of the individual.

1 According to the nomenclature of variables described in https://www.ncbi.nlm.nih.gov/projects/gap/cgi-bin/GetListOfAllObjects.cgi?study_id=phs000424.v9.p2&object_type=variable

We therefore effectively account for their influence on gene expression, mitigating these factors' impact.

This approach represents a compromise, as it is practically infeasible to ascertain the absence of underlying health conditions in the remaining samples, even if only considering cases of “fast death”. Hence, we opted to keep all samples, independently of the cause of death of its donor, to dilute potential effects associated with individual causes of death.

(2.4) The age distribution varies across tissues which may impact the results of the study. The authors' claim that age distribution does not affect the outcomes is inconclusive. Since the study aims to provide cross-tissue analysis, it is important to note that differing age distributions across tissues can influence the overall results. To address this, the authors should conduct downsampling to different age distributions across tissues and evaluate the level of tissue-specific or common changes that remain after the distributions are made similar.

We acknowledge that variations in age distributions are evident across different tissues, with brain tissues displaying a notably pronounced disparity (green density lines in Figure R2 below).

To address this issue comprehensively, we conducted tissue-specific downsampling, by reducing the number of samples in a given age window to the minimum available sample size within all age windows for a given tissue. The histograms (density plots) of the number of samples per age window of 16 years considered in the ShARP-LM model, as well as the minimum number of samples in each age window, per tissue are illustrated in Figure R1. After performing downsampling, we computed the logFC and p-value of differential expression for each gene, per age window, and compared them (for all genes in a given age window) with those involving all samples.

Despite changes in logFC with downsampling, a considerable positive correlation is maintained (Figure R3, top panel). This suggests that the overall trends in gene expression changes persist. However, the downsampling process expectedly results in a decrease of statistical power within each age window concomitant with the decreased sample size, evident from the shift of genes from the third to the first quadrant in Figure R3, bottom panel. Consequently, we have opted for maintaining results encompassing all samples and removing the paragraph in the Discussion that asserted the absence of age distribution impact on the overall outcomes (“Indeed, we found no confounding between the distribution of samples’ ages and the trend of gene expression progression over age in any tissue.”), as we deem it inaccurate, potentially leading to misinterpretation. We have added a supplementary figure (Supplementary Figure 8, identical to Figure R3) illustrating the effect of downsampling, and the following paragraph to the manuscript’s Discussion section:

“When downsampling to ensure a balanced age distribution, a loss of statistical power is apparent but a considerable positive correlation with the original results is maintained and a substantial number of significant alterations remain so (Supplementary Figure 8).”

We acknowledge that this limitation can be addressed with the growing accumulation of human tissue transcriptomes in publicly available databases, a trend we anticipate in the near future. We are committed to promptly updating voyAGEr with any new data releases that may offer a solution to this concern.

Nonetheless, we want to underscore, as the reviewer has astutely pointed out, that while voyAGEr can facilitate cross-tissue comparisons, it must be done with caution. In this regard, we inserted the following paragraph into the Discussion:

“Due to the tissue-specific nature of the pre-processing steps (v. Read count data preprocessing in the Methods section), and given that most of the plotted gene expression distributions are centred and scaled by tissue, it is important to note that voyAGEr may not be always suited for direct comparisons between different tissues. For instance, it does not allow to directly ascertain if a gene exhibits different expression levels in different tissues or if the expression of a particular gene in one tissue changes more drastically with age than in another tissue.”

(2.5) The GTEx resource is extremely valuable, however, it comes with challenges. GTEx contains tissue samples from the same individuals across different tissues, resulting in varying degrees of overlap in sample origin across tissues as not all tissues are collected for all individuals. This could affect the similar/different patterns observed across tissues. As this tool is meant for broader use by the community, it is crucial for the authors to either rule out this possibility by conducting a cross-tissue comparison using a non-parametric model that accounts for the dependency between samples from the same individual, or to provide information on the degree of similarity between samples so that the users can keep this possibility in mind when using the tool for hypothesis generation.

We agree that the variable degrees of overlap between tissues (Figure R4) could lead to a confounding between trends in a population of common individuals and those associated with age. We therefore examined the contributions of variables 'donor,' 'tissue,' and 'age' to the overall variance in the data (Figure R5, panel A), having normalised the data collectively across all tissues. Tissue and donor contribute approximately 90% and 10% of the variance, respectively. Age exhibits minimal impact (around 1%), which may be attributed to the relative subtlety of its effects on gene expression and to the tissue specificity of ageing-associated changes. Notably, removing the 'donor' variable does not transfer this variance to 'age', suggesting a limited confounding between these variables (see Figure R5, panel B).

We also specifically examined the pairs of tissues exhibiting the lowest (Brain Amygdala / Small Intestine), median (Pancreas / Heart Left Ventricle), and highest (Kidney Cortex / Muscle Skeletal) percentages of shared donors. We identified and selectively removed samples from shared donors while maintaining the original sample size imbalance between tissues. Subsequently, we calculated each gene’s mean expression within each age window from the ShARP-LM pipeline, followed by each gene’s Pearson’s correlation of expression between tissue pairs. The resulting coefficients, both with and without the removal of common donors, were compared in scatter plots (Figure R6, left plots). As this process inherently involves downsampling, which may impact results (v. comment 2.4), we performed additional downsampling by randomly removing samples from both tissues according to the proportions defined for the removal of common donors (Figure R6, right plots).

In the chosen scenarios, we note a similar impact between the targeted removal of common donors and random downsampling. Nevertheless, the effects of removing samples may vary according to the absolute number of remaining samples. Consequently, singling out individual cases may not provide conclusive insights. To systematically address this, we represented all tissue pairs in a heatmap, colour-coded based on whether the removal of common donors is more impactful (red) or less impactful (blue) than random downsampling (Figure R7). The values depicted in the heatmap, denoted as the Impact of Common Donors (ICD), are computed for each tissue pair. This calculation involves several steps: first, we determined the absolute difference in Pearson’s correlation for each gene’s mean expression within each age window from the ShARP-LM pipeline, between the original data and the subset of data without common donors (DiffWoCD) or with random downsampling (DiffRD). Subsequently, the medians of DiffWoCD and DiffRD are computed, and the difference between these median values provides the ICD for each tissue pair. Due to the unidirectional nature of correlation (i.e., the results for tissue 1 vs tissue 2 mirror those for tissue 2 vs tissue 1), the resulting matrix is triangular in form.

We have added a supplementary figure (Supplementary Figure 4, a composition of Figures R4-R7, together with a scatterplot relating the values of heatmaps R4 and R7) that aims to provide guidance to users when interpreting specific tissue pairs, acknowledging inherent limitations (refer to comment 2.4). We have also inserted the following paragraph into the manuscript’s Discussion section:

“Furthermore, we must emphasise that the majority of GTEx donors contributed samples to multiple tissues (Supplementary Figure 4A), potentially introducing biases and confounders when comparing gene expression patterns between tissues. Our analyses of variance (Supplementary Figure 4B) and downsampling to control for common donors (Supplementary Figures 4C-E) suggest very limited global confounding between the impacts of donor and age on gene expression and that any potential cross-tissue bias not to depend much on the proportion of common donors (Supplementary Figure 4E). However, this effect must be taken into account when comparing specific pairs of tissues (e.g., Colon – Transverse and Whole Blood, Supplementary Figure 4D).”

(2.6) The authors aimed to create an open-source and ever-evolving resource that could be adapted and improved with new functionality. However, this goal was only partially achieved. Although the code for the web app is open source, crucial components such as the statistical tests or the linear model are not included in the repository, limiting the tool's customizability and adaptability.

We greatly appreciate the reviewer’s concern and share their commitment to maintaining the principles of openness, reproducibility, and adaptability for voyAGEr. voyAGEr was primarily designed as a visualisation tool, displaying pre-processed results, and indeed only the code for the Shiny app itself was accessible through the project's GitHub repository.

To address this shortcoming, we have made the entire data preprocessing script publicly available in the GitHub repository of voyAGEr. This script encompasses, among others, filtration, normalisation, batch effect correction, the ShARP-LM pipeline and statistical tests employed, and module definition. Moreover, the web app itself offers functionality to export relevant plots and tables.

(2.7) Furthermore, the authors' choice of visualization platform (R shiny) may not be the best fit for extensibility and open-source collaboration, as it lacks modularity. A more suitable alternative could be production-oriented platforms such as Flask or FastAPI.

We appreciate this thoughtful concern. The decision to use Shiny was primarily driven by our data having already been prepared in the R environment during pre-processing steps. Consequently, and as the web app serves the purpose of visualisation only (and not data processing), Shiny is as a natural and convenient extension of our scripts, enabling data visualisation seamlessly.

We acknowledge that Shiny may lack the modularity required for optimal open-source collaboration. While we recognise the merits of alternative platforms like Flask or FastAPI, we decided to keep Shiny because the current iteration of voyAGEr offers significant value to the community. Transitioning to a different platform would be a time-consuming endeavour, that would postpone the release of such resource.

However, the reviewer’s feedback regarding modularity and open-source collaboration is duly noted and highly valuable. We will certainly take it into account when developing new web applications within our laboratory.

(2.8) To facilitate collaboration and improve the tool's adaptability, data resulting from the preprocessing pipeline should be made publicly available. This would make it easier for others to contribute and extend the tool's functionality, ultimately enhancing its value for the scientific community.

As outlined in point 2.6 of this rebuttal letter, certain metadata used in our analysis are subject to restricted access. To address this, we have taken several measures to foster transparency and reproducibility of our analyses. First, we have made the scripts for data pre-processing publicly available, along with a comprehensive explanation of our methodology within the main manuscript. This empowers users to replicate our analyses and provides a foundation for those interested in contributing to the tool's development. Furthermore, we have created new issues on voyAGEr’s GitHub repository, outlining novel features and improvements we envision for the application in the future. We actively encourage users to engage with this section.

(2.9) It is unfortunate that the manuscript has no line numbers, which makes pointing out language issues or typos cumbersome. Below are some minor typos present in the current version mostly due to inconsistent usage of British vs US English, and the authors would be advised to do a thorough proofreading for the final submission.Page 12: Inconsistent spelling of "analyzed" and "analysed". Should be "analyzed", since US English is used throughout the rest of the paper.Page 14: "randomised"Page 15: "emphasise"

We apologise for it and include line numbers in the revised version. We have opted for British English and corrected the manuscript accordingly.

(2.10) Some figures in the supplemental material have a low resolution (e.g. S. Fig 5). Especially figures that are not based on screenshots would ideally be of a higher resolution.

As voyAGEr is designed as a web application for visualisation, it is inherent that some screenshots of the final resource may have lower resolutions. In response to this concern, we re-generated the figures in this manuscript with a resolution that maintains clarity and readability. We also recreated figures not derived from screenshots, further improving their resolution.

We saved all figures in PDF format and are sending them together with this letter and the revised manuscript, to address any potential issues related to low-resolution figures that may occur during the export of the Word document.

<(2.11) In Fig. 1 in the bottom row the sex labels are hard to see.

We have adapted the figure to address this concern.

(2.12) Math symbols and equations are not well formatted. For example, the GE equation on p. 13, or Oiij equation should be properly typeset. Also, the Oiij notation might be confusing, I believe the authors meant to use a capital "I", i.e. OI_ij.

We have incorporated these recommendations into the revised manuscript.

(2.13) The Readme file in the git repo is very short. It would be helpful to have build and run instructions.

We have updated the README file in the GitHub repository, which now contains, among other features, instructions for launching the Shiny app and building the associated Docker image. Additionally, a simple tutorial has also been included to assist users in navigating through voyAGEr's functionalities.

(2.14) "Module" tab's UI inconsistent to other tabs (i.e. "Gene" and "Tissue"), since it contains an "About" page. Adding the "About" page in the actual "Module" page might make the UI clearer.

We believed that the Modules section, due to its distinct methodology, would benefit from an additional tab explaining its underlying rationale. We relate to the reviewer’s concern regarding the use of tabs throughout the application and made changes to the app in order to ensure consistency.

(2.15) I would suggest changing the type of the article to "Tools and Resources".

We agree and followed the reviewer’s suggestion.

**Reviewer 3**
(3.1) In the gene-centric analyses section of the result, to improve this manuscript and database, linear regression tests accounting for the entire range of age should be added. The authors' algorithm, ShARP-LM, tests locally within a 16-year window which makes it has lower power than the linear regression test with the whole ages. I suspect that the power reduction is strongly affected in the younger age range since a larger number of GTEx donors are enriched in old age. By adding the results from the lm tests, readers would gain more insight and evidence into how significantly their interest genes change with age.

We are grateful for the reviewer's thoughtful and pertinent recommendation and have thus conducted linear regression tests covering the entire age range. The outcomes of these tests have been integrated into the web application, denoted by a dotted orange line on the 'Gene Expression Alterations Over Age' plots. Additionally, a summary of statistics of overall changes, encompassing pvalues, t-statistics, and logFC per year, has been included below the plot title. We have also updated the manuscript to include such changes (v. Methods, Gene-centric visualisation of tissue-specific expression changes across age):

“We also applied a linear model across the entire age range, thereby providing users with more insight and supporting evidence into how a specific gene changes with age. For visualisation purposes, we incorporated a dashed orange line, with the logFC per year for the Age effect as slope, in the respective scatter plots (Figure 3B c). We depict the Sex effect therein by prominent dots on the average samples, with pink and blue denoting females and males, respectively.”

Concerning the observation about the potential reduction in statistical power due to the limited number of samples in younger ages, we acknowledge its validity. Indeed, we have addressed this issue in the manuscript's Discussion (v. Supplementary Figure 6).

(3.1) In line with the ShARP-LM test results, it is not clear which criterion was used to define the significant genes and the following enrichment analyses. I assume that the criterion is P < 0.05, but it should be clearly noted. Additionally, the authors should apply adjusted p-values for multiple-test correction. The ideal criterion is an adjusted P < 0.05. However, if none or only a handful of genes were found to be significant, the authors could relax the criteria, such as using a regular P < 0.01 or 0.05.

We apologise for any confusion regarding the terminology "significant genes." Our choice to use nonadjusted p-values for determining the significance of gene expression changes with Age, Sex, and their interaction was deliberate, and we would like to clarify our reasoning:

(1) In the "Gene" tab of the application, individual genes are examined. When users inquire about a specific gene, multiple-testing correction of the p-value does not apply.

(2) In the "Tissue" tab, using adjusted p-values and a threshold of 0.05 yielded very few differentially expressed genes, limiting the utility of Peaks. Our objective therein is not to assess the significance of alterations in individual genes but to provide a metric for global alterations within a tissue. We then determine significance based on the False Discovery Rate (FDR), using the p-values as a nominal metric of gene expression alterations.

To avoid using the concept of “differential expression”, commonly linked to significance, we now refer to 'altered genes' in both the manuscript and the app. For clarity and to align with voyAGEr's role as a hypothesis-generation tool, we define 'altered genes' as those with non-adjusted p-values < 0.01 or < 0.05, as discriminated in the Methods section.

(3.3) In the gene-centric analyses section, authors should provide a full list of donor conditions and a summary table of conditions as supplementary.

We appreciate the suggestion and we have now included a reference that directs readers to those data, alternatively to including this information as an additional supplementary table. We would like to emphasise that the web app includes information on donor conditions we hypothesise to affect gene expression.

(3.4) The tissue-specific assessment section has poor sub-titles. Every title has to contain information.

We agree and revised the sub-titles to more accurately reflect the information conveyed in each corresponding section.

(3.5) I have an issue understanding the meaning of NES from GSEA in the tissue-specific assessment section. The authors performed GSEA for the DEGs against the background genes ordered by tstatistics (from positive to negative) calculated from the linear model. I understand the p-value was two-tailed, which means that both positive and negative NES are meaningful as they represent up-regulated expression direction (positive coefficient) and down-regulated expression direction (negative coefficient) with age, respectively, within a window. However, in the GSEA section of Methods, authors were not fully elaborate on this directionality but stated, "The NES for each pathway was used in subsequent analyses as a metric of its over- or downrepresentation in the Peak". The authors should clearly elaborate on how to interpret the NES from their results.

We added the following paragraph to the manuscript’s Methods section, in order to clarify the NES’ directionality:

“We extracted the GSEA normalised enrichment score (NES), which represents the degree to which a certain gene set is overrepresented at the extreme ends of the ranked list of genes. A positive NES corresponds to the gene set’s overrepresentation amongst up-regulated genes within the age window, whereas a negative NES signifies its overrepresentation amongst down-regulated genes. The NES for each pathway was used in subsequent analyses as a metric of its up- or down-regulation in the Peak.”

(3.6) In the Modules of co-expressed genes section, the authors did not explain how or why they selected the four tissues: brain, skeletal muscle, heart (left ventricle), and whole blood. This should be elaborated on.

We apologise for not providing a detailed explanation for this selection. As the ‘Modules of coexpressed genes’ section was primarily intended as a proof of concept, we opted to include tissues for which we had a substantial number of samples available and availability of comprehensive cell type signatures, those being the tissues that met such criteria. Nonetheless, as the diversity of cell type signatures increases (e.g., through the increasing availability of scRNA-seq datasets), we plan to encompass a wider range of tissues in the near future. However, as this task is time-demanding and in order to avoid a substantial delay in the release of voyAGEr, we opted to approach this issue in the next version of the App and included a dedicated issue in the projects’ GitHub repository so that users can share their preferences of the next tissues to include.

We also added a brief sentence in this regard to the Methods section of the manuscript:

“The four tissues (Brain - Cortex, Muscle - Skeletal, Heart - Left Ventricle, and Whole Blood) covered by the Module section of voyAGEr were selected due to their relatively high sample sizes and availability of comprehensive cell type signatures. The increasing availability of human tissue scRNA-seq datasets (e.g., through the Human Cell Atlas) will allow future updates of voyAGEr to encompass a wider range of tissues.”

(3.7) In the modules of the co-expressed genes section, the authors did not provide an explanation of the "diseases-manual" sub-tab of the "Pathway" tab of the voyAGEr tool. It would be helpful for readers to understand how the candidate disease list was prepared and what the results represent.

We greatly appreciate the reviewer's feedback, and in response, we have restructured the 'Modules of co-expressed genes' method section to provide a more comprehensive explanation of the 'diseases' sub-section. To clarify, we obtained a curated set of diseases and their associated genes from DisGeNET v.7.0. We assessed the enrichment of modules in relation to these diseases through two methods: a manual approach utilising Fisher’s tests (i.e. comparing the genes of a given module with the genes associated with a given disease) and another through use of the disgenet2r package, employing the function disease_enrichment. Significance of these enrichments were determined by adjusting p-values using the Benjamini-Hochberg correction.

(3.8) Most figures have low resolutions, and their fonts are too small to read.

As already mentioned in issue 2.10, we have recreated all of the images with better resolution to enhance legibility. We also exported such figures in PDF, which we attach to this revision.

(3.9) Authors used GTEx V7, which is not latest version. Although researchers have developed a huge amount of pipelines and tools for their research, most of them were neglected without a single update. I am sure many users, including myself, would appreciate it if the authors kept updating the database with GTEx V8 for the future version of the database.

We express our gratitude to the reviewer for their valuable suggestion, and, as already explained in issue 2.1, we have incorporated GTEx V8 into voyAGEr.

(3.10) I would like to have an option for downloading the results as a whole for gene, tissue, and coexpressed genes. This would be a great option for secondary analysis by users.

The implementation of such feature would be a time-demanding endeavour that would delay the release of voyAGEr, and we therefore chose not to perform it for this version. However, we agree that it would be a good resource for secondary analyses and acknowledge the possibility of adding this feature in the future. For now, voyAGEr allows the user to download all plots and corresponding data.

(3.11) How the orders of tissues in the heatmaps (both gene and tissue section) were determined? Did the authors apply hierarchical clustering? If not, I would recommend the authors perform the hierarchical clustering and add it to display the heatmap display.

We apologise for the oversight in explaining the process behind determining the order of tissues. To clarify, we employed hierarchical clustering to establish the tissue order for visualisation within the app. Although the reviewer suggested adding a dendrogram to illustrate this clustering, we decided against it. The reason for such is that including a dendrogram, while informative, is not essential for the app's primary purpose.

(3.12) I understand that this is a vast amount of work, but I hope that the authors can expand the coexpressed module analysis to include other tissues in the future version of the database.

Knowing what co-expressed genes in line with aging are and their pathway and disease enrichments across tissues would be highly informative, and I'm sure many users, including myself, would greatly appreciate it.

We express our gratitude to the reviewer for the valuable suggestion and for acknowledging the extensive effort required to incorporate new tissues into the module section. We completely agree that understanding co-expressed genes across the aging process is of significant value, and we are committed to the ongoing inclusion of additional tissues. As already stated in issue 3.6, comprehensive list of tissues slated for integration in future voyAGEr versions is readily available on voyAGEr’s GitHub repository.

**Author response image 1. sa4fig1:** Density plots (“smoothed” histograms) of the distribution of numbers of samples per moving age window for the ShARP-LM pipeline, categorised by tissue. The numerical value within each rectangle represents the minimum number of samples observed across all age windows for that particular tissue.

**Author response image 2. sa4fig2:** Density lines (“smoothed” histograms) of the distribution of the age of donors per tissue. As depicted in the chart, there are more samples for older ages, particularly of brain tissues.

**Author response image 3. sa4fig3:** Effect of downsampling in ShARP-LM results. (**A**) Per tissue violin plots of gene-wide distributions of Pearson’s correlation coefficients between original and downsampled logFC values for the Age variable across age windows, with tissues coloured by and ordered by increasing percentage of downsampling-associated reduction in the number of samples. (**B**) – Density scatter plots of comparison of associated original and downsampled p-values for each tissue, coloured by the downsampling percentage in each age window, highlighting the low range of p-values (from 0 to 0.1). Despite changes in logFC with downsampling, a considerable correlation in significance is maintained, although downsampling naturally results in a loss of statistical power, evident by the shift of points towards the first quadrant (dashed lines: p-value = 0.05).

**Author response image 4. sa4fig4:** Heatmap depicting the percentage of common donors between pairs of tissues. A given square illustrates the percentage of all samples of tissue in the x axis (Tissue 1) that is in common with the tissue in the y axis (Tissue 2)

**Author response image 5. sa4fig5:** Assessment of the relative contributions of different sources to the dataset’s variance. (**A**) - tissue accounts for approximately 90% of the total variance, while donor contributes around 10%; age has a minimal impact (1%), likely due to the relative subtlety of its effects on gene expression and to the tissue specificity of ageing dynamics. (**B**) - Removal of the donor variable does not transfer variance to age, suggesting limited confounding between the two variables.

**Author response image 6. sa4fig6:** Impact of the relative proportion of common donors on gene expression correlation between tissue pairs. Panels A, B, and C showcase the tissue pairs with the highest (Muscle Skeletal / Kidney Cortex), median (Pancreas / Heart Left Ventricle), and lowest (Small Intestine / Brain Amygdala) percentages of common donors, respectively. The left panels illustrate gene-bygene Pearson’s correlations of gene expression between the two tissues, comparing the scenarios with (x-axis) and without (yaxis) the removal of common donors. The ri ght panels depict the same comparisons, but with random downsampling (y-axis) in both tissues based on the proportions defined for common donor removal. The depicted examples show that the outcomes are comparable when removing common donors or employing random downsampling.

**Author response image 7. sa4fig7:** Comparison of the impacts of removing common donor samples and random downsampling across tissue pairs. The heatmap is coloured based on whether the removal of common donors has a greater (red) or lesser impact (blue) than random downsampling. The values depicted in the heatmap, denoted as the Impact of Common Donors (ICD), are computed for each tissue pair. This calculation involves several steps: first, by determining the absolute difference in Pearson’s correlation for each gene’s mean expression within each age window from the ShARP-LM pipeline, between the original data and the subset of data without common donors (DiffWoCD) or with random downsampling (DiffRD). Subsequently, the medians of DiffWoCD and DiffRD are computed, and the difference between these median values provides the ICD for each tissue pair. Due to the unidirectional nature of correlation (i.e. the results for tissue 1 vs tissue 2 mirror those for tissue 2 vs tissue 1), the resulting matrix is triangular in form. Gray tiles denote NA values, i.e., where the tissue-tissue comparison does not have a meaning, namely self-self and between sex-specific tissues. Top right insert: density line (“smoothed” histogram) of all ICD values.